# Antioxidant, Bacteriostatic and Preservative Effects of Extractable Condensed Tannins Isolated from Longan Pericarps and Seeds

**DOI:** 10.3390/plants12030512

**Published:** 2023-01-22

**Authors:** Mengli Wang, Ting Chen, Qin Wang, Yan Shi

**Affiliations:** 1School of Life Sciences, Xiamen University, Xiamen 361102, China; 2Université de Paris, CiTCoM-UMR 8038 CNRS, U 1268 INSERM, F-75006 Paris, France; 3National Demonstration Center for Experimental Life Sciences Education, Xiamen University, Xiamen 361102, China

**Keywords:** longan pericarp and seed, extractable condensed tannins, antioxidant capacity, food preservation, fresh-cut lotus roots

## Abstract

In the process of longan production and processing, a large amount of remnants is produced, such as dried longan pericarps and seeds, which have been reported to be rich in polyphenols but not effectively utilized. In this paper, the total phenolic contents in the remnants of longan pericarps and seeds were found to be 39.58 ± 3.54 and 69.53 ± 1.99 mg/g (DW), respectively, accounting for 60–80% of those in fresh samples. The contents of extractable condensed tannins (ECTs) in the remnants of longan pericarps and seeds were 19.25 ± 6.71 mg/g (DW) and 44.59 ± 2.05 mg/g (DW), respectively, accounting for 60–70% of the fresh samples. These data indicate that the polyphenols in the remnants of the sampled longan pericarps and seeds were effectively retained. The antioxidant capacity of ECTs from the longan pericarps and seeds was more than 60% of the fresh samples measured with the 1, 1-diphenyl-2-trinitrophenylhydrazine and ferric reducing ability of plasma methods. Further exploration showed that ECTs from the longan pericarps and seeds had significant inhibitory effects on *Pseudomonas aeruginosa*, *Escherichia coli*, *Salmonella* and *Staphylococcus aureus*. The minimum inhibitory concentration (MIC) of the longan pericarp ECTs on all four studied bacteria was 3 mg/mL. The MIC of longan seed ECTs on Salmonella was 3 mg/mL, and that of the other three bacteria was 1.5 mg/mL. In view of the good antioxidant and antibacterial activities of longan pericarps and seeds, we applied them to the preservation of fresh-cut lotus roots. When the concentration of ECTs in the longan pericarps and seeds was 2 mg/mL and 1 mg/mL, respectively, the two kinds of ECTs showed an obvious preservative effect. After the ECT treatment of the lotus roots, their browning degree was reduced, their color was better maintained, their respiration was inhibited and their nutrient loss was reduced. Bacterial reproduction was inhibited, and cell senescence was slowed. Accordingly, the shelf life of ECT-treated fruits and vegetables can be effectively extended. Overall, we can suggest that ECTs from the remnants of dried longan pericarps and seeds could be used as natural preservatives for fresh-cut fruits and vegetables.

## 1. Introduction

Dimocarpus longgana Lour. (longan) is a famous subtropical fruit in China. It has been cultivated for more than 2000 years and is known as “one of the four best fruits in Lingnan” [1]. Longan can not only be eaten fresh as fruit but can also be processed into dried longan, a popular traditional Chinese food that is easy to store. Longan pericarps and seeds (which account for 17% and 15% of the whole fresh fruit weight, respectively) can also be used as a kind of Chinese traditional medicine [2]. In dried longan processing, tens of thousands of dried longan pericarps and seeds are discarded every year, which causes environmental pollution and is a waste of resources. Therefore, the reuse of dried longan pericarps and seeds has great development value.

Studies have indicated that longan pericarps and seeds are rich in polyphenols and other active substances that have antioxidation, anticancer and antiaging effects [3]. Plant polyphenols are the fourth most common natural product in nature after cellulose, hemicellulose and lignin [4]. Polyphenols are also known as tannins, including hydrolyzed tannins and extractable condensed tannins (ECTs), which are secondary metabolites produced by plants to resist climatic variations. In 1990, low-molecular-weight gallic acid and tea polyphenols were listed in GB2760–86 (Chinese Standards for Food Additives) as new natural nontoxic food additives due to their excellent antioxidant activity [5]. Therefore, plant polyphenols are also called natural antioxidants [6,7]. In addition, Prasad et al. [8,9] found that polyphenols extracted from lychee seeds and pericarps could effectively inhibit tyrosinase activity. Tyrosinase (EC 1.14.18.1) is an enzyme involved in melanin production, and tyrosinase inhibitors may be clinically useful for the treatment of skin cancer. Reddy et al. [10] extracted hydrolyzed tannins from pomegranate pericarps and combined them with penicillin to inhibit the growth of *Pseudomonas aeruginosa*, *Cryptococcus neoformans* and *Glucose aureus*. In recent years, an increasing number of studies have focused on the extraction and function of longan polyphenols. Yuttana Sudjaroen et al. extracted ellagic tannins from longan polyphenols and found that they had antioxidant effects [11]. Zhuo Wang et al. measured the contents of polyphenols and analyzed the antioxidant activity of eight longan varieties in China, finding that these varieties were all rich in polyphenols with antioxidant activity [12]. Ting He et al. found that longan seed polyphenols could interact with starch, which is expected to regulate the content of starch in blood glucose [13]. Moreover, some scholars revealed the molecular basis of the polyphenols and pathogen resistance of longans with whole-genome analysis [14], which provided molecular evidence for the preservation value of longans.

With the continuous improvement in people’s living standards and the acceleration in the pace of life, ready−to−eat foods such as fresh-cut fruits and vegetables have become a consumption trend [15,16]. However, fresh-cut fruits and vegetables physically produced by pruning, peeling or cutting are more susceptible to deterioration than whole fruits and vegetables [17,18]. Therefore, prolonging the shelf life of fresh-cut fruits and vegetables is a key problem in the fresh-cut fruit and vegetable industry [19,20].

In view of the above research background, we extracted extractable condensed tannins (ECTs) from the pericarps and seeds of fresh and dried longans, and we compared their total phenolic contents and the antioxidant activities. The results show that the ECTs were better retained in the dried longan pericarps and seeds. They demonstrated a strong effect when we applied them in fresh-cut lotus root preservation, which suggested that ECTs in dried longans can be reused as promising preservation agents for fresh-cut fruits and vegetables.

## 2. Results and Discussion

### 2.1. Total Phenolics and Extractable Condensed Tannins of Longan Pericarps and Seeds

The total phenolic (TP) and extractable condensed tannins (ECTs) contents were determined, and the results are shown in Table 1.

As shown in Table 1, the total phenolic contents of the fresh longan pericarp (FLP) and fresh longan seed (FLS) were 57.48 ± 2.81 mg/g (DW) and 78.82 ± 1.35 mg/g (DW), respectively. The total phenolic contents in the dried longan pericarp (DLP) and dried longan seed (DLS) were 39.58 ± 3.54 mg/g (DW) and 69.53 ± 1.99 mg/g (DW), respectively, accounting for 68.86% and 88.21% of the fresh samples, respectively. The ECT contents in the FLP and FLS were 33.52 ± 5.23 mg/g (DW) and 56.67 ± 3.29 mg/g (DW), respectively. The ECT contents were 19.25 ± 6.71 mg/g (DW) and 44.59 ± 2.05 mg/g (DW) in the DLP and DLS, respectively, accounting for 57.43% and 78.68% of the fresh samples, respectively. Based on these data, the total phenolic and ECT contents in the longan seeds were higher than those in the longan pericarps, whether in fresh or dried samples. In addition, based on the analysis of the proportion of the total phenolic and ECTs contents in the dried longans, the loss of the total phenolic and ECTs contents in longan seeds during processing was less than that in longan pericarps. The reason for these results may have been that the longan pericarp has a large contact area and is a thin layer, while the longan seed is a solid ball that can better ensure the safety of its internal structure. In brief, a dried longan retains a large nutritional portion and can be reused.

### 2.2. Antioxidant Activity of Longan Pericarps and Seeds

In the determination of the 1, 1-diphenyl-2-trinitrophenylhydrazine (DPPH) free radical scavenging rate and the ferric reducing ability of plasma (FRAP) total reducing ability, the fresh/dried longan pericarps, longan seeds and positive control vitamin C showed significant dose effects. As shown in Figure 1A,B, the DPPH clearance rate of longan seed ECTs was higher than that of longan pericarp ECTs in both fresh and dried samples. A comparison between fresh/dried longan pericarps and seeds showed that the DPPH clearance rates of the dried samples were lower than those of fresh samples, which was consistent with the total phenolic and ECTs contents measured in the samples. As shown in Figure 1C,D, the total reduction capacity measured with FRAP was consistent with that measured with DPPH.

The IC50 value represents the concentration when the inhibition effect reaches 50%. The smaller the IC50 value is, the stronger the free radical scavenging effect is. Therefore, the IC50 value can be used to effectively measure the antioxidant capacity of different samples. In this study, FRAP was measured as the standard equivalent of positive control vitamin C (mg AAE/mg), as shown in Table 2.

The IC50 values of the DPPH clearance rate of the DLP and DLS were 264.80 ± 5.45 and 167.11 ± 2.59 μg/mL, respectively, which were 18.6% and 22.8% higher than that of the fresh samples. Regarding FRAP values, there were no significant differences between the DLS and FLS, but the ECTs activity of the DLP decreased by 27.7% compared with the FLP. The ECTs in longan seeds mainly consist of hydrolyzed tannins, including gallic acid, ellagic acid and corilagin, and the antioxidant activity of hydrolyzed tannins is higher than that in condensed tannins [21]. In this study, the ECTs content and antioxidant activity of the longan seeds were higher than those of the longan pericarps, which may have been related to the contents of hydrolyzed tannins; however, further experiments are needed to confirm this idea. In conclusion, the antioxidant activity of ECTs from the dried longan pericarps and seeds accounted for more than 70% of the fresh samples, thus demonstrating the value of resource reuse.

### 2.3. Analysis of Bacteriostatic Ability

#### 2.3.1. Analysis of Inhibition Ability to Common Bacteria

*Pseudomonas aeruginosa* (a), *Escherichia coli* (b), *Salmonella* (c) and *Staphylococcus aureus* (d) were selected to detect the inhibitory effect of ECTs from the DLP and DLS on bacteria.

In this section, we examined the antibacterial efficiency of the dried samples since the total phenolic contents of the fresh and dried samples were different and had no comparative value. As is clearly seen in Figure 2, significant antibacterial rings appeared in holes with the DLP and DLS ECTs, and the dose effect was obvious following increases in the ECT concentration.

On the basis of the experimental results, the ECT concentrations were further diluted to 0.375, 0.75, 1.5, 3 and 6 mg/mL, and the other experimental operations were the same as those in the experiments described above. The minimum inhibitory concentration (MIC) of different plant polyphenol samples to bacteria was determined by observing whether there was a bacteriostatic zone, and the results are shown in Table 3.

The minimum inhibitory concentration (MIC) of the ECTs from the DLP to the four kinds of bacteria was 3 mg/mL; the MIC of the ECTs from the DLS to Salmonella was 3 mg/mL, and the MIC of the ECTs from the DLS to the other three kinds of bacteria was 1.5 mg/mL. At a concentration of 1.5 mg/mL, the ECTs in the longan seeds showed an inhibitory effect on some bacteria, but the ECTs in the longan pericarps had no inhibitory effect on these bacteria. Some studies have reported that tannins have the ability to inhibit the growth and protease activity of ruminal bacteria by binding the cell walls of bacteria [22,23]. Nuchanart Rangkadilok et al. reported the antifungal activities of longan seed extracts, but they concluded that none of the longan extracts showed an inhibitory effect on the tested bacteria strains at a concentration of 1000 μg/mL [24]. Based on previous research, we tested the antibiotic activity of ECTs, and our result suggested that 1.5 mg/mL of ECTs from longan seeds showed antibacterial activity, which was consistent with the literature.

#### 2.3.2. Total Bacterial Count of Fresh-Cut Lotus Roots

Fresh-cut fruits and vegetables are more conducive to microbial growth because of the seepage of nutrients caused by cellular structure damage. In this study, the total number of bacteria in fresh-cut lotus roots showed a growth trend with increases in storage time (Figure 3). Furthermore, the total number of bacteria in the control group was significantly greater than that in the experimental group. The total number of bacteria in the ECTs of the DLP treatment group was significantly lower than that in the ECTs of the DLS treatment group (*p* < 0.05). On the 4th day in the control group, the total number of bacteria reached 872.50 ± 45.96 × 10^3^ cfu g^−1^, while the experimental group had almost one−tenth as many colonies as the control group and this trend had become more pronounced through time. Bacteria and fungi are the main causes of the decay of fruits and vegetables [25], and ECTs can significantly reduce the colony content, indicating a remarkable preservation function.

### 2.4. Preservation Effect on Fresh-Cut Lotus Roots

#### 2.4.1. Apparent Changes in Fresh-Cut Lotus Roots

The photos in Figure 4 show the apparent changes in the fresh-cut lotus root slices during storage. Starting from the 4th day, obvious browning appeared in the control group slices, small black spots appeared on the edges of the DLP treatment group slices, and the DLS treatment group slices maintained a good state. On the 6th day, the DLP treatment group slices were significantly darker than before, but the DLS treatment group slices remained in good condition. From the 8th day, all treatment group slices showed some black edge spots, but the DLS treatment group slices were in the best condition. By the 12th day, lotus roots in the control group were completely black and odorous. The experimental group results were significantly better than the control group results, and the DLS treatment lotus roots were in better condition than the DLP treatment lotus roots. The results showed that the DLP and DLS ECTs had strong preservative effects on the fresh-cut lotus roots, with the DLS ECTs showing comparatively better effects.

#### 2.4.2. Browning Degree of Fresh-Cut Lotus Roots

The browning degree of fresh-cut lotus roots was evaluated by detecting the changes in the surface color difference and absorbance values of a water solution [26]. The color difference values are shown in Figure 5A. It was found that the L* value decreased with increases in storage time. After being treated and left at room temperature for 2 h, the L* value of the experimental group was significantly higher than that of the control group (*p* < 0.05), which indicated that the fresh-cut lotus roots showed an antibrowning effect after the DLP and DLS ECT treatments in a short time. On the 2nd day, the L* of all lotus roots significantly decreased due to the physical damage of the fresh-cut lotus roots, which led to the release of ECT content. Based on the literature, we inferred that the REDOX reaction rate increased two days before storage, and a large amount of melanin was produced [27]. On the 4th day, the L* value of the control group continued to decrease, but the treatment group maintained a strong state, especially the DLS treatment group, whose L* value showed little difference from that of the 2nd day. After the 6th day, there were significant differences between the three groups. The effect of the DLS treatment group was significantly better than that of the DLP treatment group, and the effect of the DLP treatment group was better than that of the control group (*p* < 0.05).

As a lotus root slice has several holes, some small areas could not be detected with a chromometer. As a supplement, the absorption value of the fresh-cut lotus roots at 420 nm was detected with the extinction coefficient method. The extinction coefficient method can be used to directly detect the soluble pigment compounds produced in the storage process of fresh-cut lotus roots. The detection results are shown in Figure 5B. Consistent with the L* results, the browning degree of the control group was significantly higher than that of the experimental group (*p* < 0.05). In the course of storage, this gap was gradually amplified, while the fresh-cut lotus roots treated with longan pericarp and longan seed ECTs maintained their initial state. The browning degree of the experimental group began to increase from the 2nd day, and this rate gradually increased. In contrast, the browning degree of the fresh-cut lotus roots in the experimental group showed almost no change in the first 4 days, remaining at a low level of about 0.045. From the 6th day, the browning degree of the longan pericarp ECTs treatment group slightly increased to 0.167 while that of the longan seed ECTs treatment group was still lower than 0.05, which was significantly lower than that of the longan pericarp group (*p* < 0.05). On the 10th day, the browning degree of the experimental groups significantly increased to about 0.4 and 0.5 in the longan seed and longan pericarp ECTs treatment groups, respectively. According to the reported browning values, the longan pericarp and seed ECTs were able to maintain the fresh-cut lotus roots for about 10 days.

#### 2.4.3. Total Phenolic and Soluble Quinone Contents in Fresh-Cut Lotus Roots

Phenolics are the substrates of enzymatic browning, and soluble quinones are the intermediate products of enzymatic browning. They have multiple physiological and functional activities in plants [28]. Figure 5C shows that the TP content showed a trend of first decreasing and then increasing. This may have been related to the activity of polyphenol oxidase (PPO). Studies have shown that the PPO activity of fresh-cut fruits and vegetables first sharply increases and then slowly decreases during storage time, but it is generally higher than that of intact fruits and vegetables [29]. When plant tissues are in a harsh environment or pathological state, PPO activity is increased to enhance plant resistance, which may be the reason for the decrease in the TP content observed in this study over the first two days. On the 4th day, the TP content in the control group was higher than that in the experimental group (*p* < 0.05) and the PPO content continued to accumulate. These results may indicate that severe browning occurred in the experimental group on the 4th day because the substrate sharply grew and led to browning. On the 6th day, the TP content in the experimental group showed a small peak and the fresh-cut lotus roots in the experimental group showed significant browning. From the 6th to 12th days, the TP content decreased, which may have been caused by the slowing of various metabolic processes due to the aging of the fresh-cut lotus root tissue.

Compared with the total phenolic content, the soluble quinone content increased with the increase in storage time, which means that the browning was continuous. As can be seen in Figure 5D, the soluble quinone content of the experimental group was lower than that of the control group, demonstrating that ECTs treatment of longan pericarps and seeds could effectively inhibit the browning of fresh-cut lotus roots; there were no significant changes between the pericarp and seed groups. The changes in the total phenolic and soluble quinone contents further verified the L* value and browning degree results.

#### 2.4.4. Weight Loss Rate of Fresh-Cut Lotus Roots

The weight loss curve is shown in Figure 5E. From the 2nd to 12th days, the weight loss rate of the control group was significantly higher than that of the experimental group. From the 4th to 8th days, the weight loss rate of the control group was almost twice that of the longan pericarp and seed ECTs treatment groups. On the 12th day, the weight loss rate was 19.07 ± 1.25% in the control group, which was three times higher than that in the longan pericarp ECTs treatment group (6.64 ± 1.63%) and the longan seed ECTs treatment group (7.36 ± 0.80%). The effect of the longan seed ECTs on the weight loss rate was slightly lower than that of the longan pericarp ECTs, but the difference was not significant.

#### 2.4.5. Malondialdehyde Content in Fresh-Cut Lotus Roots

MDA is the main product of membrane lipid peroxidation under the action of lipid oxidase and can be used to evaluate the degree of membrane peroxidation: the more the MDA content, the more serious the damage and aging of cells. The change in the MDA content of the fresh-cut lotus roots was detected with the extinction coefficient method, as shown in Figure 5F. The MDA content of the fresh-cut lotus roots fluctuated during the whole storage period, which may have been related to the activities of lipoxidase. However, the MDA content in the control group was higher than that in the experimental group (*p* < 0.05). Further analysis showed that the MDA content in the control group sharply increased to 0.75 µmol/g on the second day compared with 0.1 µmol/g in the experimental group. These results indicate that in the first two days of storage, cell membrane lipids in the control lotus slices were greatly oxidized, resulting in cell damage and the accumulation of MDA. The MDA content in the experimental group increased on the 2nd and 6th days; however, the increases were small, which indicated that the ECTs in longan pericarps and seeds could effectively delay the senescence process of fresh-cut lotus root cells.

## 3. Materials and Methods

### 3.1. Experimental Materials

Fresh longan and lotus roots were bought from Meijiayuan Supermarket at the east gate of Xiamen University. Dried longan pericarps and seeds came from the remnants of fresh longans after dry processing. P. aeruginosa (Strain NO: FJAT−341), Staphylococcus aureus (Strain NO: FJAT−12029), Escherichia coli (Strain No: Fjat−7239) and Salmonella typhimurium (strain NO. Fjat-10334) were obtained from the Fujian Academy of Agricultural Sciences; 1, 1-diphenyl-2-trinitrophenylhydrazine (DPPH), gallic acid, vitamin C and 2,4,6-tripyridyl triacridine (TPTZ) were purchased from Sigma−Aldrich. Folin−Ciocalteu reagent was purchased from Beijing Biodee Biotechnology. Peptone and yeast extracts were bought from the Oxoid company. Penicillin/kanamycin was purchased from GIBCO, and other reagents were purchased from the Sinopharm group. The water used in this experiment was purified with a Millipore Milli−Q apparatus (TG 110 Water Systems, Indianapolis, IN, USA).

### 3.2. Extraction of ECTs

For fresh longan pericarps and seeds, fresh longans of full and similar size with no insect damage were selected. The longans’ pulp was discarded, and the pericarps and seeds were obtained. Then, they were cleaned with running water and deionized water. The dried longan pericarps and seeds were then dried, ground, screened and stored as powder at −80 °C in darkness for later use. Polyphenols were extracted with the method established by Chai Weiming et al. [30]. First, 20 g of a lyophilized sample was accurately weighed, and 200 mL of a 70% (*v*/*v*) acetone aqueous solution was added at a ratio of 1:10. After the mixture was evenly stirred with a glass rod, the samples were placed in an ultrasonicator and oscillated for 30 min. The samples were filtered with a Buchner funnel, and the filter residue was extracted again according to the above steps that were repeated three times. Then, we discarded the filtered residue and merged the filtrate. Acetone was removed via the rotary evaporation of the filtrate (the temperature was not higher than 45 °C) to obtain an aqueous extract of plant polyphenols. The aqueous extract was successively extracted with petroleum ether and ethyl acetate, and each solvent was extracted three times. Petroleum ether was used to remove the lipids and pigments in the aqueous extracts, and ethyl acetate was used to remove the small phenolic molecules in the water extract. After extraction, the organic solvent and most of the water were again removed with rotary evaporation, and a white or reddish-brown powder, which was the crude extract of the plant polyphenols, was obtained after freeze-drying.

The crude polyphenol extract was further purified with column chromatography, with Sephadex LH−20 glucose gel as the stationary phase. The crude polyphenol extract was dissolved in a 50% (*v*/*v*) methanol solution, and the sample was taken according to the size of the gel column (the gel column was first balanced with 50% methanol). After all the samples flowed into the column, the impurities in the samples were eluted with a 50% methanol aqueous solution, and then the samples were collected with 70% (*v*/*v*) acetone. The purified plant ECT samples were obtained via rotating evaporation and freeze-drying [21].

### 3.3. Determination of Total Phenolic Content

The total phenolic (TP) content was determined via the Folin−Ciocalteu (FC) method [31]. The specific operation process was as follows: First, 0.2 mL of an aqueous sample was measured, and then 0.3 mL of sterile water and 0.5 mL of the FC reagent were successively added and mixed. Next, a 2.5 mL sodium carbonate solution (20%) was added to the mixture and placed at room temperature for 40 min, and the absorbance value was measured at 725 nm. Water was used as the negative control, and gallic acid was used as the standard.

The gallic acid standard curve was formulated as follows: We accurately prepared the gallic acid aqueous solution at concentrations of 40, 80, 120, 160 and 200 μg/mL, and then we measured the absorbance value according to the FC method. The standard curve was obtained by plotting the concentration of gallic acid and the absorbance value, and then we calculated the TP content with the standard curve. The unit of the TP content is mg GAE/g DW.

### 3.4. Determination of ECT Content

The content of soluble extractable condensed tannins (ECTs) was determined via the BuOH−HCI method established by Terrill et al. [32]. The detailed procedure is as follows [33]: A 1 mL sample aqueous solution was added to a 6 mL *n*-butanol/hydrochloric acid (95:5, *v*/*v*) solution, which was gently shaken and placed in a boiling water bath for 75 min. After the reaction liquid was cooled, the absorbance value was measured at 550 nm. The negative control was water, and the standard comprised purified plant-extractable condensed tannins.

To formulate the standard curve of purified plant tannins, we first prepared a purified plant−extractable condensed tannin sample solution with gradients of 0, 20, 40, 60, 80 and 100 µg/mL. Then, we measured the absorbance value according to the BuOH-HCI method. The standard curve was obtained by plotting the concentration of the standard substance with the absorption value. The unit of ECTs content is mg/g.

### 3.5. Antioxidant Capacity of ECTs

#### 3.5.1. DPPH Method

We determined the free radical clearance rate with the DPPH method [34]. A DPPH solution was prepared with methanol at a concentration of 0.004% (25 μg/mL). Plant ECTs samples were diluted with the methanol solution at different concentration gradients (15.63, 31.25, 62.5, 125 and 250 μg/mL). The methanol solution was used as the negative control, and vitamin C was used as the standard.

The reaction steps were as follows: A 0.1 mL sample or standard solution was added to a 3 mL DPPH solution; this mixture was fully shaken and kept standing at room temperature for 30 min in the dark, and the absorbance value was measured at 517 nm. The results are expressed as IC50 (μg/mL). The DPPH radical scavenging rate = [(A0 − A1)/A0] × 100, where A0 is the absorbance value of the negative control and A1 is the absorbance value of the sample/standard product.

#### 3.5.2. FRAP Method

The antioxidant capacity of ECTs was determined with the FRAP method according to the work of Benzie and Strain et al. [35]. Plant ECT samples were diluted with water to prepare a series of concentrations. Water was used as the negative control, and vitamin C was used as the standard. A 0.1 mL sample or a standard solution with different concentrations was mixed with 3 mL of the FRAP reagent. The mixed solution was then placed in a 25 °C water bath for 10 min, and the absorbance value was measured at 593 nm. Results are expressed as mg ascorbic acid per mg sample (mg AAE/mg)

### 3.6. Antibacterial Activity of ECTs

#### 3.6.1. Bacterial Culture

The *Pseudomonas aeruginosa*, *Staphylococcus aureus*, *Escherichia coli* and *Salmonella typhimurium* strains were taken from the refrigerator at −80 °C and thawed at room temperature. We added 0.1 mL of each strain to a 5 mL sterilized LB medium. We then incubated them in a shaker (37 °C and 150 rpm) and activated them for 8 h. Then, a 1% concentration of the activated bacteria solution was cultured in a medium for 12 h at 37 °C and 150 rpm on a rotary shaker.

#### 3.6.2. Bacteriostatic Activity

The antibacterial activity was evaluated via the Oxford cup method based on the work of Liu Dongmei et al. [36]. First, 10 mL of a 1.5% LB solid medium was poured into a Petri dish before solidification, and the Oxford cup was appropriately placed. The bacteria liquid and the upper medium were mixed at a ratio of 1:15 (*v*/*v*) and slowly poured into the Petri dish. After the upper medium solidified, the Oxford cup was removed. Then, a 50 μL ECTs solution with different concentrations was added to each well before being placed in a CO_2_ incubator (37 °C and 12 h). Sterile water was used as the negative control, kanamycin was used as the positive control for Pseudomonas aeruginosa, and ampicillin was used as the positive control for the other bacteria. We set two of each type of bacteria in parallel. An ECTs aqueous solution inhibits bacterial growth and produces an inhibitory ring, so the lowest ECTs concentration that could produce an inhibitory ring in the experiment was the determined MIC value (mg/mL) of the bacteria.

#### 3.6.3. Total Colony Count

The total number of bacteria was found in accordance with GB/T5009−96 “Food Hygiene Inspection Method” via the spread plate method. We precooled lotus roots at 4 °C for 24 h; then, we peeled and evenly cut them into slices with a thickness of 4 mm. ECTs concentrations of 2 mg/mL in the DLP group and 1 mg/mL in the DLS group were prepared. The slices of fresh-cut lotus root were randomly and evenly distributed, and then they were soaked at room temperature for 20 min. Under sterile conditions, the fresh-cut lotus roots were ground with sterile water in ratios of 1:10, 1:100, or more. The LB medium, cooled to about 50 °C, was poured into the plate. After the medium solidified, 0.1 mL of each dilution sample was taken into the sterile Petri dish and coated with a stick. Three dilution samples were created in parallel and placed in an incubator at 36 ± 1 °C for 24 ± 1 h for colony counting. The results were equal to the total colonies multiplied by the dilution ratio of the samples (cfu·g^−1^), and aseptic water was used as the negative control. Results were recorded every two days.

### 3.7. Effects of ECTs on Preservation of Fresh-Cut Lotus Roots

The lotus roots that were precooled at 4 °C for 24 h were taken out, peeled and evenly cut into slices with a thickness of 4 mm. ECT concentrations of 2 mg/mL from the longan pericarps and 1 mg/mL from the longan seeds were prepared, and aseptic water was used as the negative control. The slices of fresh lotus roots were randomly and evenly distributed, and then they were soaked at room temperature for 20 min. The slices were taken out, drained and placed at room temperature for 2 h. The first data were recorded on day 0. Then, fresh-cut lotus roots in different treatments were packed into PE bags and placed in a refrigerator at 4 °C. Results were recorded every two days.

#### 3.7.1. Apparent Changes in Lotus Roots

After the lotus roots were soaked with different solutions, photos were taken every two days to record apparent changes.

#### 3.7.2. Color Change and Browning Index

The experimental method was based on the work of [37], and color changes were measured using an automatic colorimeter (Chentaike, Beijing, China). The L*, a* and b* values (where L* represents brightness, a* and b* represent chromaticity coordinates, +a* represents red, −a* represents green, +b* represents yellow and −b* represents blue) of the fresh-cut lotus roots were measured every two days. During detection, three points were randomly selected for each lotus root piece, and 12 points were measured in a group, which was repeated three times in parallel.

The browning index was measured via the extinction coefficient method [38]. First, 40 mL of water and 10 mL of a 10% (*v*/*v*) trichloroacetic acid aqueous solution were added to 2 g samples and ground into homogenates. The homogenates were placed at 35 °C for 2 h and then filtered. The absorbance values were measured at a wavelength of 420 nm.

#### 3.7.3. Total Phenolic and Soluble Quinone Contents

Referring to the method of Gao et al. [39], 5 g of lotus roots treated with different solutions was weighed. Then, 20 mL of methanol was added, ground into homogenate and centrifuged at 5000 rpm for 10 min. The supernatant was collected as a crude extract for reserve. The total phenolic content was determined by using the Folin phenol method [40]. We added 0.2 mL of a crude extract to 1 mL of a Folin–Ciocalteu reagent and 3 mL of a Na_2_CO_3_ reagent. Next, we added water to constant volume of 10 mL, and then left the mixture at room temperature for 1 h. The absorbance value was measured at 760 nm. Gallic acid was used as the standard, and the results are expressed as gallic acid equivalents.

The soluble quinone content was determined according to the work of [41]. A crude extract was prepared with methanol, and then the absorbance value was measured at 437 nm. The results are expressed as OD437 g^−1^.

#### 3.7.4. Determination of Weight Loss Rate

Weight loss rate (%) = (the weight before storage − the weight after storage)/weight before storage × 100%.

#### 3.7.5. Malondialdehyde Content

The method of Dhindsa et al. [42] was used for the determination of malondialdehyde content.

First, 10 mL of 10% trichloroacetic acid (containing 0.5% 2-thiobarbituric acid) was added to 2 g of lotus roots, which were ground into a homogenate. The mixture was brought to a boil and heated for 10 min. After being cooled, it was centrifuged at 12,000 rpm for 10 min. Then, we took out the supernatant, and we measured the absorbance values at 450, 532 and 600 nm. The malondialdehyde (MDA) content (µmol/g) = 6.45 × (OD532 − OD600) − 0.56 × OD450.

### 3.8. Statistical Analysis

Three parallels were set each time, and the results are expressed as mean ± SD in the IBM SPSS Statistics 20.0 program. Statistical comparisons were performed with a one-way analysis of variance (ANOVA). In the results, “a” represents the reference group, b represents significant differences with the “a” group (such as *p* < 0.05), c represents more significant differences with the “a” group (such as *p* < 0.01), and so on.

## 4. Conclusions

We extracted plant polyphenols from food processing scraps (dried longan pericarps and seeds). Then, we tested the biological activities of polyphenols and applied them to the preservation of fresh-cut lotus roots. Our experimental results show that dried longan pericarps and seeds have good recycling value. Compared with those of the fresh longans, the total phenolic and extractable condensed tannin contents of dried longan pericarps and seeds were well reserved. The results obtained using the DPPH and FRAP methods show that the ECTs in the longan pericarps and seeds had antioxidant effects. We also found that the longan pericarp and seed ECTs had a strong antibacterial effect. After fresh-cut lotus roots were treated with dried longan ECT solutions, the L* value of the lotus roots in the experimental group slowly decreased and their browning degree decreased, maintaining good sensory qualities. Changes in the weight loss rate were small, the peak of the CO_2_ respiration rate was delayed, and the respiration rate decreased, which reduced the consumption of nutrients. The increase rate of MDA slowed and the total number of bacteria exponentially decreased, which prevented cell senescence and the production of harmful substances. In general, ECTs from longan pericarps and seeds can effectively slow the spoilage process and prolong the shelf life of fresh-cut lotus roots. Plant polyphenols are kinds of mixtures with similar properties, so it is difficult to separate and purify them. We will try to purify our polyphenols and perform structural identification in the future. The mechanism of fresh-cut lotus root preservation and the variable trend of related enzymes have not been explored, so we will conduct further research in this field.

## Figures and Tables

**Figure 1 plants-12-00512-f001:**
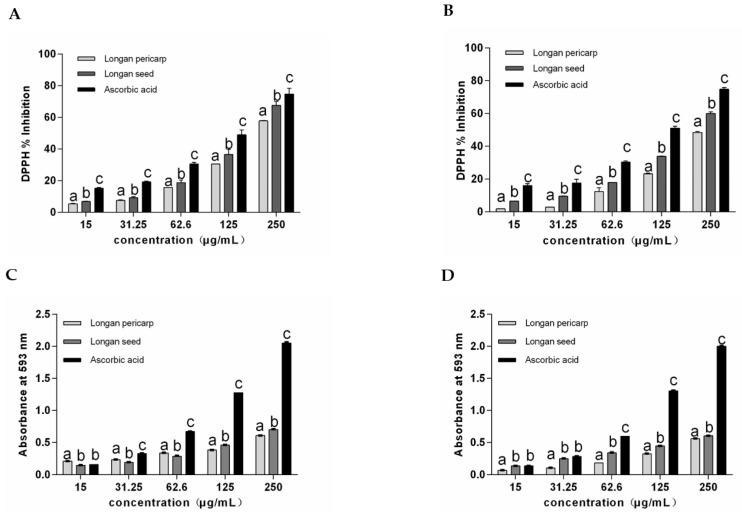
Antioxidant capacity of ECTs in longan pericarps and seeds: (**A**) DPPH values of FLP and FLS; (**B**) DPPH values of DLP and DLS; (**C**) FRAP values of FLP and FLS; (**D**) FRAP values of DLP and DLS. In the results, “a” represents the reference group, b represents significant differences with the “a” group (such as *p* < 0.05), c represents more significant differences with the “a” group (such as *p* < 0.01).

**Figure 2 plants-12-00512-f002:**
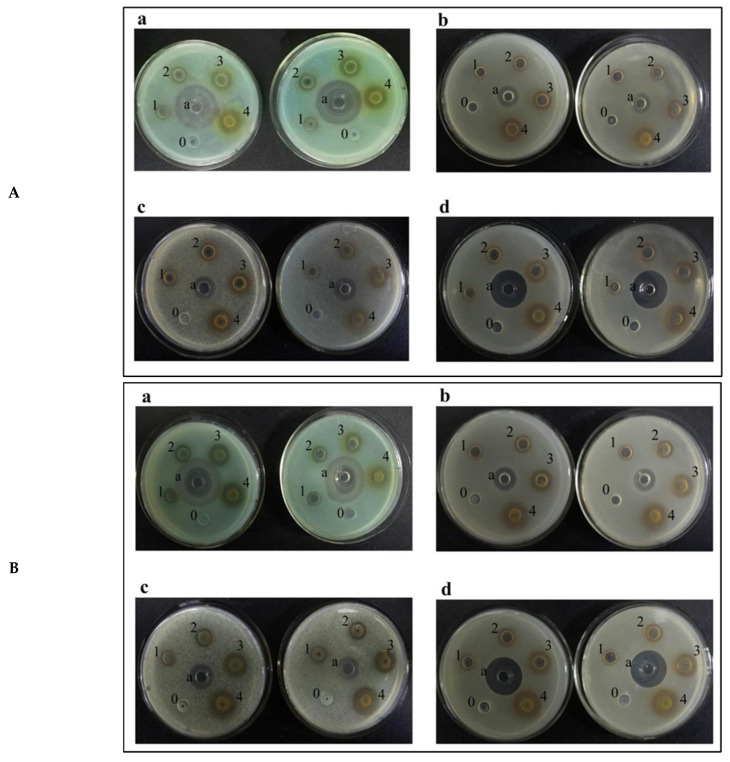
The antibacterial effect of ECTs from the DLP (**A**) and DLS (**B**): (**a**) *Pseudomonas aeruginosa*, (**b**) *Escherichia coli*, (**c**) *Salmonella* and (**d**) *Staphylococcus aureus*. Wells 0, 1, 2, 3 and 4 represent ECT concentrations of 0, 3.125, 6.25, 12.5 and 25 mg/mL, respectively. The middle wells are the positive control represented by a.

**Figure 3 plants-12-00512-f003:**
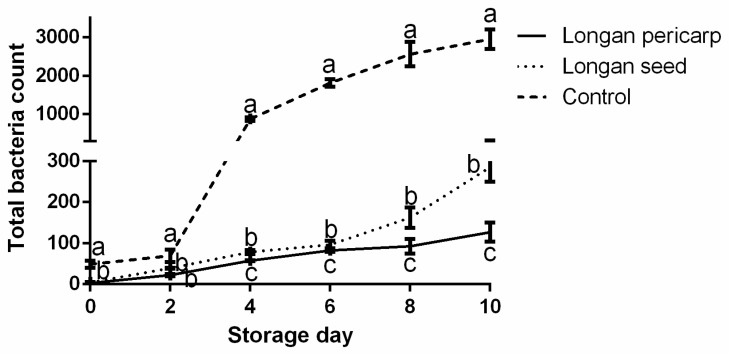
Changes in the total bacteria contents of fresh-cut lotus roots. The units in the x− and y−axes denote the day and cfu g^−1^, respectively. In the results, “a” represents the reference group, b represents significant differences with the “a” group (such as *p* < 0.05), c represents more significant differences with the “a” group (such as *p* < 0.01).

**Figure 4 plants-12-00512-f004:**
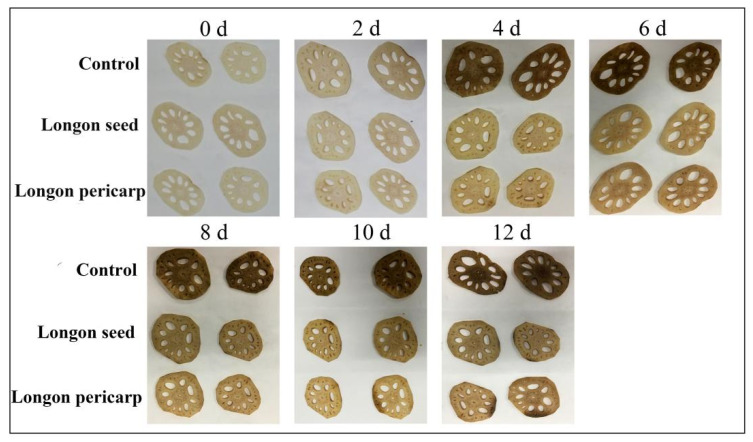
Apparent changes in fresh-cut lotus roots treated with different solutions during storage for 12 days at 4 °C.

**Figure 5 plants-12-00512-f005:**
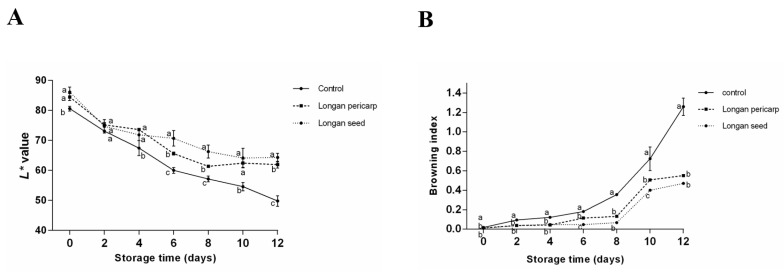
The preservation effect of fresh-cut lotus roots after being treated with different solutions during storage for 12 days at 4 °C: (**A**) changes in color values; (**B**) changes in the browning index; (**C**) changes in the total phenolic contents; (**D**) changes in the soluble quinone contents; (**E**) changes in weight loss (%); (**F**) changes in the MDA content. Letters indicate statistical significance. In the results, “a” represents the reference group, b represents significant differences with the “a” group (such as *p* < 0.05), c represents more significant differences with the “a” group (such as *p* < 0.01).

**Table 1 plants-12-00512-t001:** TP and extractable condensed tannin contents in longan pericarps and seeds.

Samples	Total Phenolics (mg/g)	Extractable Condensed Tannins (mg/g)
Fresh longan pericarp (FLP)	57.48 ± 2.81	33.52 ± 5.23
Fresh longan seed (FLS)	78.82 ± 1.35	56.67 ± 3.29
Dried longan pericarp (DLP)	39.58 ± 3.54	19.25 ± 6.71
Dried longan seed (DLS)	69.53 ± 1.99	44.59 ± 2.05

**Table 2 plants-12-00512-t002:** Antioxidant activities of longan ECTs.

	DPPH (IC50 μg/mL)	FRAP (mg AAE/mg)
Fresh longan pericarp	215.67 ± 1.24	0.47 ± 0.20
Fresh longan seed	167.11 ± 2.59	0.64 ± 0.29
Dried longan pericarp	264.80 ± 5.45	0.34 ± 0.10
Dried longan seed	198.42 ± 3.76	0.62 ± 0.31
Vitamin C	112.80 ± 1.65	1

**Table 3 plants-12-00512-t003:** The antibacterial effect of polyphenols extracted from longan pericarps (A) and seeds (B).

Concentration of ECTs (mg/mL)										
Strain	6		3		1.5		0.75		0.3625	
	A	B	A	B	A	B	A	B	A	B
a	1.1	1.13	0.93	1.05	/	0.9	/	/	/	/
b	1.03	1.08	0.95	0.98	/	0.95	/	/	/	/
c	1.03	1.13	0.95	1.05	/	/	/	/	/	/
d	1.05	1.18	0.98	1.02	/	0.9	/	/	/	/

a *Pseudomonas aeruginosa*, b *Escherichia coli*, c *Salmonella* and d *Staphylococcus aureus*. The data represent the diameter of the bacteriostatic zone (cm).

## Data Availability

Data openly available in a public repository.

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
