# Peer review of "Antioxidant, Bacteriostatic and Preservative Effects of Extractable Condensed Tannins Isolated from Longan Pericarps and Seeds"

_plants, 2023, doi:10.3390/plants12030512_

Round 1

Reviewer 1 Report (Previous Reviewer 1)

The current version of the resubmitted manuscript is significantly improved compared to the previous one. Most of the statements that I had found confusing or misleading are corrected or removed from the text. However, I believe, the Authors should pay attention to some issues before the manuscript can be considered for publication. The detailed remarks are presented in the attached file.

Author Response

Reviewer 2 Report (New Reviewer)

The manuscript entitled, “Biological activity of extractable condensed tannins from longan pericarp and seed and their effect on fresh-cut lotus roots can benefit the researchers working in the area of post-harvest technology and microbiology. However, the manuscript needs some work before being consided for publication in Plants. Therefore, major revision is recommended concerning the quality of draft.

The suggestions are as follows:

The title doesn’t really justify the manuscript, consider changing it to ‘Antioxidant, bacteriostatic and preservative effects of extractable condensed tannins isolated from longan pericarp and seed’

Abstract

Line 11-13: The sentence is grammatically incorrect, please rephrase.

Line 17-18: Remove ‘(DPPH)’ and ‘(FRAP)’ as it appears only once throughout the abstract.

Line 25: What do you mean by fresh-keeping effect? Better to use a scientific term.

Line 38: Expected? The sentence seems inappropriate, if it is not backed by experiments. Better rephrase or delete it.

Line 29-30: Change the last sentence ‘These results suggested that ECTs in the remains of dried longan pericarp and seed can be reused as promising preservation agents for fresh-cut fruits and vegetables.’

To

‘Overall, from the outcome we can suggest that ECTs from the remains of dried longan pericarp and seed has prospects to be used as a natural preservative for fresh-cut fruits and vegetables.’

Introduction

Line 35: Why is L capital in Longgana?

Line 37: Replace ‘it’ with ‘is’

Line 37-39: Break the sentence in two and add reference for replenishing blood.

Line 43-44: Sentence seems incomplete and incorrect, please rephrase!

Line 45-46: Rephrase the following sentence and cite a reference, ‘Although longan pericarp and seed have some medicinal history in Chinese folk, there are relatively few reports about the functions of longan pericarp and seed.’

Line 48-52: Break the sentence into two and reword ‘changeable external environment’ with ‘climatic variations’.

Line 56: What is the relevance here for tyrosinase activity inhibition? Add a meaning to this statement with addition information.

Line 59-60: Rephrase!

Line 64: What is meant by ‘different antioxidant activity’

Line 66: ‘What’s more’? Rephrase this, its better to stick to scientific language.

Line 67: But what was the outcome of the study?

Materials and methods

Line 360: Delete ‘was based on the method’ and add a recent reference for DPPH assay. Consider https://doi.org/10.1007/s40089-020-00320-y

Line 379-383: The entire section should be rewritten with scientific language. What is meant by ‘put’, it should be ‘incubated’.  

Line 380: ‘Dissolved’? What was used to dissolve them?

Line 459: Abrupt start, please rephrase.

The material and methods section needs some work, particularly with regards to English and grammer.

Results and Discussion

Section 2.3.1.: Elaborate on the discussion part, stating why the antibacterial potential was tested and how it will benefit the field of antibiotic resistance. Consider https://doi.org/10.1016/j.heliyon.2019.e02021 for discussion, and https://www.mdpi.com/2223-7747/6/2/16 to pick some examples.   

Discussion can be improved significantly.

 Conclusion

Add an introductory statement to the conclusion, it starts abruptly.

Authors should also identify and comment on at least a couple of future directions to their findings.

Round 2

Reviewer 1 Report (Previous Reviewer 1)

The revised manuscript is, in my opinion, almost ready for further publication procedure. Regarding my previous review (point 6a referring to different ECTs concentrations), I only ask the Authors to check and clarify the phrase “Referring to the method of Gao et al. [38], 5 g of lotus roots treated with different ECT solutions was weighed.” (lines 469-470). What does “different ECT solutions” mean?

Author Response

Reviewer 2 Report (New Reviewer)

Authors have adequately addressed the previously provided comments. However, some grammatical errors and spacing issues persists, for example on line 371 there should be spacing between the numbers -  40,80,120,160,200 μg/mL.

Also, authors should add a recent reference in section 3.3, for instance, consider https://doi.org/10.1016/j.ijbiomac.2022.06.145. The manuscript can be accepted after these minor amendments, and suggestions from other reviewers and editor are addressed. I thank the authors for their efforts towards improving the manuscript.  

Author Response

This manuscript is a resubmission of an earlier submission. The following is a list of the peer review reports and author responses from that submission.

Round 1

Reviewer 1 Report

In my opinion, the manuscript needs much improvement before it can be considered for publication in the journal. There are many results of observation included in the text, but no deeper analysis and conclusions are presented, only the results of observations are described. Moreover, the plot of the manuscript is unclear and it is hard to follow the text, especially due to ununiform and unclear language and phrases and statements throughout the text. Therefore, I recommend deep revision of the manucript and resubmittin it after the improvement is completed. In the attached file I present my comments and remarks that may help in improving the quality and clearance of the manuscript.

Reviewer 2 Report

 The present manuscript was aimed to investigate the polyphenols in the pericarp and seed extracts of longan and their uses. The total phenolic contents and soluble condensed tannins were quantified by spectrophotometric methods, and their antioxidant bioactivities were determined by DPPH and FRAP assays. In addition, the inhibitory effects of these polyphenols against several bacteria were examined. According to these data, authors performed the preservation study of fresh cut lotus root. Although I appreciated authors' efforts, this manuscript displayed various technical deficiencies. The present results were preliminary and would not show any significant impacts to the natural products related research fields. In brief, this manuscript is not recommended to accept for publication in Plants. In addition, there are several major comments to be addressed as following.

1.         The manuscript was written well, however, there were still some minor typographical, grammar and format errors presented. For example, the names of bacteria in abstract should be in italics. Authors have to check and revise these errors carefully.

2.         The experimental results provided by colorimetric method were not so interesting since there were a lot of interferences influencing the obtained data. It meant that the determination of total phenolics and condensed tannins may not be contributed by the indicated compounds only. These contents should be at least determined with some standards by HPLC before resubmission.

3.         The antioxidant activities were only examined in the extracts level and no any molecular data were incorporated. In addition, the antioxidant potentials were not significant as compared with the positive control. The acceptance of this manuscript should be reconsidered only after the examination of single compound was really performed.

4.         In the antibacterial examination, the concentration of positive control should be noted in the Figure and Experimental section. Similarly, the antibacterial potentials were not significant as compared with the positive control.

5.         In the References section, the writing manner of several references did not follow the style of this journal. Authors have to check and revise these errors carefully.

Reviewer 3 Report

In this study, authors determined the biological activity of polyphenol from longan pericarp and seed and their effect on fresh-cut lotus roots. The paper is well written, but minor revision before publishing (listed in later text) is needed. All technical errors need to be corrected.

1)     It should be emphasized that the mass concentration is indicated on the x-axis

2)  Exponent when displaying the number of bacteria should be in superscript (line 185)

3)     In many cases there is no space between the variable and the measuring unit (For example in lines 208, 422…). Please go through the entire manuscript in detail and correct this.

4)     In Na2CO3 (line 426) and CO2 (line 456) numbers should be put in subscript